# OpenReview forum: "The Elicitation Game: Evaluating Capability Elicitation Techniques"
_ICML.cc/2025/Conference — ICML 2025 poster_

### Official Review · Reviewer_CMEX · 2025-02-17

**Overall Recommendation:** 5

**Summary:**

This paper aims to test capability elicitation techniques for evaluating language models. The motivation is that models are often deployed with undiscovered harmful capabilities that later resurface in deployment. Building on prior work for password-locked models, they introduce models with more sneaky passwords that are harder to elicit with simple prompting techniques. They do this by using 'circuit breaking' to suppress the backdoor behavior unless the password is present. They test 9 capability elicitation techniques based on password-locked and circuit-breaking password-locked models. The key finding of the paper is that fine-tuning methods are distinctly the most effective at elicitating latent capabilities.

**Claims And Evidence:**

I feel convinced by this paper (and related work) of how fine-tuning methods are particularly effective at capability elcitation. This provides a useful and compelling piece of evidence for this. I think that this work did what it set out to do very well.

The main thing that this paper does not do is show that its model organisms are particularly good models of real-world capabilities. One could red-team models that had undergone unlearning or refusal fine-tuning. This would have been nice but related things have already been done, and not doing this doesn't negate the clear value of this paper.

**Essential References Not Discussed:**

Could the authors work on adding some more discussion of backdoors (e.g., https://ieeexplore.ieee.org/abstract/document/9996741?casa_token=I1LX2IcCjbMAAAAA:sin7jUwhWhI5aqHqeqws2uilBXiCuUCa6BPaafVNxBu2Oz1wCiEvPVp8HY1WFTY0IUtofkCS4w) and model tampering attacks (e.g., https://arxiv.org/abs/2502.05209)?

**Experimental Designs Or Analyses:**

The key experimental appraoch involved training mistral and gemma models with conventional and enhanced password locking and then attacking them to re-elicit the suppressed capability. This is a sound approach.

**Methods And Evaluation Criteria:**

I found them compelling. One could always complain about using more models/tasks, but I think that the experimental scope was reasonable.

**Other Comments Or Suggestions:**

None

**Other Strengths And Weaknesses:**

Strength: I might be biased because I am familiar with the subject matter, but this paper was easy to read and understand.

Weakness: This paper did an adequate but not particularly thorough job of discussing related work. In particular, it seems important to discuss more about backdoors (including obfuscated backdoors) to credit past work related to this.

**Questions For Authors:**

I thinkt the future work paragraph is reasonable, but FWIW, I think that bridging findings from model organisms with real model evals seems like a good priority.

**Relation To Broader Scientific Literature:**

I work on some closely-related topics, and I think that the results in this paper are not totally unexpected. Nonetheless, I strongly believe that this work is useful and citeable.

**Theoretical Claims:**

No theory (not needed).

---

> ### Author Rebuttal · Authors · 2025-04-01
>
> Thanks for your help and feedback! We’re glad that you found the motivation and contributions compelling.
>
> > The main thing that this paper does not do is show that its model organisms are particularly good models of real-world capabilities. One could red-team models that had undergone unlearning or refusal fine-tuning. This would have been nice but related things have already been done, and not doing this doesn't negate the clear value of this paper.
>
> Do you agree with our response to Tqzj that:
>
> “This is a great point and something that we gave some consideration! We think that our model organisms provide an upper-bound for the difficulty of eliciting hidden capabilities from real systems. This is because our models are not constrained to generate realistic behaviour hiding capabilities, e.g., the circuit-broken models can output nonsensical tokens in the coding setting, whereas real sandbagging models would likely need to sandbag in an undetectable way, e.g., by imitating the behaviour of a weaker system. This is why we say in the paper that our model organisms “act as a worst-case, adversarial scenario” for eliciting methods. From a practical standpoint, it seems plausible that elicitation methods which are effective on our model organisms would also be effective on real systems (though the converse is less likely). We will add further discussion of this point to the paper.”
>
> > I found them compelling. One could always complain about using more models/tasks, but I think that the experimental scope was reasonable.
> We’re glad that you found the experimental scope compelling — Reviewers XiFg and BBji commented asking for further tasks, so we extended our results to MMLU (see response to XiFg).
>
> Thank you for providing the references to backdoors and model tampering attacks. They are definitely related to our work and we have added further discussions on them, inline with your suggestion.
>
> > I thinkt the future work paragraph is reasonable, but FWIW, I think that bridging findings from model organisms with real model evals seems like a good priority.
>
> Creating realistic model organisms is difficult, which is a main reason for why we opted for the approach described above. However, there are some exciting possibilities here, and we agree that this is a good priority.
> First, one could create model organisms with certain properties in toy scenarios and testing them in realistic scenarios (see [Denison et al., 2024](https://arxiv.org/abs/2406.10162)). Second, one could create scenarios where AI systems “decide by themselves” to hide their capabilities, and then apply elicitation techniques (see e.g., [Meinke et al., 2025](https://arxiv.org/abs/2412.04984) or [Greenblatt et al., 2024](https://arxiv.org/abs/2412.14093)).
>
> ---
> Thank you for your review!

---

### Official Review · Reviewer_BBji · 2025-03-11

**Overall Recommendation:** 2

**Summary:**

This paper evaluates techniques for eliciting hidden capabilities in language models. The authors create two types of "model organisms" (password-locked and circuit-broken models) with intentionally hidden capabilities, then test various elicitation methods including prompting, activation steering, and fine-tuning. Their findings suggest that for MCQ tasks, combined prompting techniques work well, while for code generation, only fine-tuning reliably elicits capabilities from the more robust circuit-broken models.

**Claims And Evidence:**

- The core claim that circuit-broken models are more resistant to elicitation than password-locked models is supported, particularly for code generation.
- The evidence showing the effectiveness of combining techniques is clear and convincing for these specific tasks/domains.

**Essential References Not Discussed:**

--

**Experimental Designs Or Analyses:**

- The experimental methodology is generally sound
- However, very limited scope of testing - only two tasks/domains with mostly saturated benchmarks
- Would have benefited from testing more diverse elicitation in domains with real-world impact. The authors even mention that there being only
four reasonable tokens (A, B, C, or D) for WMDP, the elicitation method merely needs to guide the output to one correct token --> this is certainly not the case if these elicitation and model organisms techniques are to be applied/studied more broadly.

**Methods And Evaluation Criteria:**

- The model organism approach is nice, but a rehashed idea that is evaluated in this work
- Evaluation on multiple-choice (WMDP) and code-generation (APPS) provides decent coverage for small-scale study purpose
- I also wonder whether the relatively small-scale models used (7B-9B parameters) are representative of frontier systems where capability hiding is a real concern and the domains and style of tasks used. How does this change with model size? How does this change with reasoning models trained with rewards?

**Other Comments Or Suggestions:**

--

**Other Strengths And Weaknesses:**

Strengths:
- Practical insights for AI evaluators on some methods to prioritize going forward modulo the tasks chosen
- Clear and interpretable presentation of results

Weaknesses:
- The paper seemed to make recommendations with limited evidence in 2 quite artificial/niche settings
- Lacks deeper analysis of why certain elicitation techniques succeed or fail. Would have been nice to deeply analyze response patterns of models qualitatively.
- The overall contribution feels a bit incremental with minimal/surface level insights into model behaviour

**Questions For Authors:**

--

**Relation To Broader Scientific Literature:**

The paper clearly builds on model organism and capability elicitation techniques from prior work, and focusses on evaluation only.

**Theoretical Claims:**

No formal proofs to check as the paper is primarily empirical.

---

> ### Author Rebuttal · Authors · 2025-04-01
>
> Thanks for your feedback. We're glad you appreciated our novel model organism approach, clear results, and practical insights.
>
> > The model organism approach is nice, but a rehashed idea that is evaluated in this work
>
> Our paper focuses on using model organisms to evaluate techniques to elicit hidden capabilities, improving past work by introducing a more robust model organism, and providing an extensive evaluation of different elicitation techniques. We note that other reviewers found our contribution compelling.
>
> > [on small scale & reasoning models, and domain/style of tasks]
>
> We used open-weight models to enable us to create model organisms and to evaluate elicitation methods requiring weight-access, and we used smaller models to meet our engineering and compute constraints. Are there specific experimental results you would like validated on larger models?
>
> We note that scaling experiments with reasoning models like DeepSeek R1 (released just days before ICML submission) would be interesting but outside our current scope. (R1 was the most prominent open-weight reasoning model at the time).
>
> > The paper clearly builds on model organism and capability elicitation techniques from prior work, and focusses on evaluation only.
>
> This is incorrect because we introduce a novel model organism, and show that it improves prior work (i.e. more robust to elicitation). We note that the other reviewers commented positively on this contribution.
>
> > However, very limited scope of testing - only two tasks/domains with mostly saturated benchmarks
>
> We've extended our results to MMLU (see response to XiFg). With WMDP, APPS, and MMLU, we cover diverse domains where hidden capabilities could be harmful.
>
> Benchmark saturation is not a concern for our study. We selected WMDP, MMLU, and APPS to focus on domains where hidden capabilities could be harmful. The importance of detecting concealed capabilities remains regardless of base model performance, and our evaluated models do not reach saturation on these benchmarks.
>
> > The authors even mention that there being only four reasonable tokens (A, B, C, or D) for WMDP [...] this is certainly not the case if these elicitation techniques are to be applied more broadly.
>
> This limitation of WMDP is exactly why we also study the (more realistic and open-ended) code-generation setting on APPS. We extended our results to MMLU, requiring the model to perform across a wide range of diverse domains to hide its capability. It also makes certain elicitation methods, such as persona or concept methods, more challenging because it is more difficult for evaluators to determine broad-covering concepts or personas that would simply “guide the output to one correct token”.
>
> > [...] recommendations with limited evidence in 2 quite artificial/niche settings
>
> As noted, the APPS code-gen setting is quite realistic, and we now extend our results to MMLU (see response to XiFg). But indeed, we invite future work on additional settings to make our recommendations more robust.
>
> >Lacks deeper analysis [...]
>
> We agree this would be valuable but providing rigorous analyses across multiple methods requires effort beyond our current scope, especially given the multiple elicitation methods. We do offer preliminary hypotheses (lines 360-372, right column) as a stepping stone for future theoretical research.
>
> An important insight from our work is that hidden capabilities can be embedded within specific layers of a model through mechanisms such as LoRA adapters and/or circuit-breaking. This suggests that evaluators who fine-tune models at specific layers—common practice with LoRA—may remain vulnerable to the persistence of hidden capabilities. Concretely, circuit-broken model organisms can be trained in a way that fine-tuning only on the latest 40% of layers does not elicit performance. While this point was accidentally excluded from the original submission, it has now been incorporated into the revised manuscript.
>
> >The overall contribution feels a bit incremental [...]
>
> To re-emphasize our contributions, we:
> - Introduce a novel model organism combining password-locking with circuit-breaking that demonstrably improves robustness to capability elicitation.
> - Provide a comparative evaluation of multiple elicitation techniques across different tasks and models, showing that fine-tuning methods (especially anti-refusal training) are most effective.
> - Demonstrate that combining techniques (N-shot, prefilling, and steering) significantly improves elicitation effectiveness when fine-tuning access is unavailable.
>
> It seems like you may have under-appreciated the novel model organism.
>
> ---
>
> Thanks again for taking the time to review our paper!
>
> We hope that the additional experiments and these responses provided help bring the paper in line with your expectations for a pioneering study of capability elicitation methods. We would be glad to address any further questions or concerns you may have.

---

### Official Review · Reviewer_Tqzj · 2025-03-13

**Overall Recommendation:** 3

**Summary:**

In this paper, the authors seek to evaluate different capability elicitation methods from the literature. Since there are too many to reasonably evaluate, they choose a subset of techniques by prioritizing their cost, simplicity and ease of use. The authors propose to use *model organisms* that mimic real-world models whose capabilities have been hidden, to evaluate the different techniques.

Motivated by the finding that password-locked models often used in the literature are brittle in the face of capability elicitation interventions, the authors combine this with the circuit-breaking training techniques, leading to more robust model organisms.
The authors then benchmark their selected elicitation techniques on their 2 model organisms (password-locked, circuit-broken), on a multiple-choice question-answering task and a coding task, on 3 models in the 7b-9b range (one base, 2 instruct).

They find that:
* their circuit-broken model is much more robust to elicitation;
* elicitation technique efficiency varies between task and model but that supervised fine-tuning is usually the best performing one;
* combining techniques performs better;
* on the more complex coding task only finetuning (supervised as well as anti-refusal) recovers the full model capability.

**Claims And Evidence:**

The claims are substantiated by evidence from 2 tasks and 3 models. I find the quality of evidence relatively satisfactory, but would be happy to see confirmation from other tasks since some results differ importantly between the MCQA and coding task. For instance, that might help understand why prefilling is so effective for coding and password-locked models (Sec 4.2), and tease out why elicitation in general is much less effective for coding (Sec 5).

**Essential References Not Discussed:**

None come to mind.

**Experimental Designs Or Analyses:**

See Methods and Evaluation section.

**Methods And Evaluation Criteria:**

I wonder whether the Mistral model makes for an accurate model organism. More recent models have undergone much more extensive post-training, which might affect how they behave under capability elicitation.

Coding is a good proxy for complex capabilities that one might want to hide. MCQA is probably slightly too simplistic for a model of dangerous capabilities as relates to misuse risks. One might expect bad actors to query the model in natural language, such that a generative evaluation of a model's harmful knowledge might be useful.

**Other Comments Or Suggestions:**

N/A

**Other Strengths And Weaknesses:**

* I am very concerned about a major limitation of this paper, namely the relationship between model organisms such as the authors study here and realistic models whose capabilities have been hidden. While the authors acknowledge the limitation themselves (this is good!), there is no discussion on how model organisms and realistic models are related experimentally. Given how variable the results are from one model organism, base model, and task, one can rightfully ask what lessons are to be drawn from this paper from a practical standpoint (except that combining elicitation techniques is better, which is unsurprizing). Since it might be hard to find realistic models whose creators have admitted they hid a capability, a study on the appropriateness of model organisms could be carried on harmfulness and then tentatively transferred to capabilities. Good discussion on this topic, or experimental evidence from the authors or from other papers could make me raise my score.
* The paper is extremely clear and very pleasant to read. The authors surveyed the literature correctly and summarized all their techniques with an appropriate level of detail and clear, to-the-point language.
* The paper could almost read as the presentation of a new model organism, by combining previous methods. The authors do show that the circuit-breaking model is more robust, and since the elicitation techniques are reasonable this makes for a stronger framing.
* “We show that anti-refusal training elicits hidden code-generation capabilities. Notably, this method does not require any code-related training data.”. This is a very interesting result and would warrant further investigation.

**Questions For Authors:**

* Question 1: Why the choice of mistral 7b instead of more recent models such as llama or qwen?
* Question 2: “supervised fine-tuning elicits the capability and often even improves performance (for the circuit-broken Gemma instruct even a 6% improvement).” (sec 4.1): As a point of reference, how does the improvement compare to fine-tuning the original Gemma instruct model?
* Question 3: Do you have any idea why the capabilities of mistral base are systematically higher than the ones of mistral instruct?

**Relation To Broader Scientific Literature:**

The authors attempt to summarize previous research in capability elicitation through this study. They cite multiple techniques, from which they select 6 with various levels of ease of application, keeping in mind constraints third-party model evaluators might face.
The model organism the authors propose to study is a combination of the password-locked model, which is classical in the elicitation literature, with a training technique coming from adversarial robustness called circuit-breaking.

**Theoretical Claims:**

N/A

---

> ### Author Rebuttal · Authors · 2025-04-01
>
> Thanks for taking the time to review our paper and offer high-quality feedback! We’re glad that you found the paper clear and enjoyable to read, and that you appreciated the strength of our novel model organism based on circuit-breaking.
>
> Note: we abbreviated some of your comments to meet the character limit.
>
> > The claims are substantiated [...] but would be happy to see confirmation from other tasks [...]
>
> As discussed in response to XiFg, recreating our results on MMLU has provided further evidence for the core claims in the paper. Experiments in non-MCQA settings is a great area for future research.
>
> For instance, that might help understand why prefilling is so effective for coding and password-locked models (Sec 4.2), and tease out why elicitation in general is much less effective for coding (Sec 5).
>
> Prefilling has been found to be quite an effective jailbreak method (e.g. [Andriushchenko et al., 2024](https://arxiv.org/pdf/2404.02151) because it causes the model to quite strongly condition its response on the prefilled tokens (e.g., "Sure here's how I can help")
>
> Elicitation is likely substantially harder for the open-ended coding setting because the "target space" is much larger, i.e., in MCQA the elicitation only needs to guide the output to a single token ("A", "B", etc), whereas the space of capable outputs in coding is much richer and difficult to elicit (see lines 360-272, right column). We invite further work to examine this and related hypotheses in more detail.
>
> > I am very concerned about a major limitation of this paper [...]
>
> This is a great point and something that we gave some consideration! We think that our model organisms provide an upper-bound for the difficulty of eliciting hidden capabilities from real systems. This is because our models are not constrained to generate realistic behaviour hiding capabilities, e.g., the circuit-broken models can output nonsensical tokens in the coding setting, whereas real sandbagging models would likely need to sandbag in an undetectable way, e.g., by imitating the behaviour of a weaker system. This is why we say in the paper that our model organisms “act as a worst-case, adversarial scenario” for eliciting methods. From a practical standpoint, it seems plausible that elicitation methods which are effective on our model organisms would also be effective on real systems (though the converse is less likely). We will add further discussion of this point to the paper.
>
> Adding more context still, we agree that creating realistic model organisms is difficult, which is a main reason for why we opted for the approach described above. However, there are some exciting possibilities here. First, one could create model organisms with certain properties in toy scenarios and testing them in realistic scenarios (see [Denison et al., 2024](https://arxiv.org/abs/2406.10162)). Second, one could create scenarios where AI systems “decide by themselves” to hide their capabilities, and then apply elicitation techniques (see e.g., [Meinke et al., 2025](https://arxiv.org/abs/2412.04984) or [Greenblatt et al., 2024](https://arxiv.org/abs/2412.14093)).
>
> > The paper could almost read as the presentation of a new model organism [and] this makes for a stronger framing.
>
> Thanks! We will try to emphasise this contribution more.
>
> > [The success of anti-refusal training] is a very interesting result and would warrant further investigation.
>
> We agree, and we are coordinating with others on follow-up work.
>
> > Question 1
>
> We chose Mistral-7b v3 because it was released 10 months ago, roughly around when we started on the project, and it achieved SOTA on various benchmarks at the time. For Gemma 2 9B instruct, we specifically selected it for the code generation tasks due to its superior performance compared to the other 7B models on this benchmark.
>
> > Question 2
>
> |Model|Setting|WMDP(%)|MMLU(%)|
> |-|-|-|-|
> |*Mistral-base*|SFT original|60.0|63.6|
> ||PW-locked + PW|58.4|62.0|
> ||Circuit-broken + PW|58.7|62.8|
> ||SFT PW-locked|60.0|66.0|
> ||SFT circuit-broken|60.0|64.0|
> |*Mistral-instruct*|SFT original|51.4|59.1|
> ||PW-locked + PW|57.3|59.4|
> ||Circuit-broken + PW|56.8|59.3|
> ||SFT PW-locked|57.0|64.0|
> ||SFT circuit-broken|53.0|61.0|
> |*Gemma-instruct*|SFT original|54.9|75.3|
> ||PW-locked + PW|52.2|72.5|
> ||Circuit-broken + PW|52.4|72.3|
> ||SFT PW-locked|58.0|**76.0**|
> ||SFT circuit-broken|**62.0**|**76.0**|
> *PW - shorthand for password*
>
> > Question 3
>
> We do not have a well-grounded explanation for this phenomenon. However, throughout our experiments, we observed that smaller instruct/chat models generally tend to perform slightly worse on MCQA tasks.
>
>
> ---
> Thanks again for your detailed feedback!
> We hope that the additional experiments and these responses provided help bring the paper in line with your expectations for a pioneering study of capability elicitation methods. We would be glad to address any further questions or concerns you may have.

---

> > ### Comment · Reviewer_Tqzj · 2025-04-06
> >
> > In light of the discussion with the other reviewers and of your answer to my comments I am happy to raise my score.
> >
> > I am not sure I completely agree with your justification that elicitation difficulty for the model organisms you introduced are an upper bound for sandbagging models: it is hard to make such claims a priori for LLMs. Your argument ("our models are not constrained to generate realistic behaviour hiding capabilities") implies that model organisms are easier to train than sandbagging models, not necessarily that they are more robust. I could imagine a genuinely misaligned model attempting to hide its capabilities being much harder to crack than a model of a similar size having undergone much shallower password-locking finetuning.
> >
> > However, if you include the above discussion in the paper, I would like to see it published, to encourage discussion and further experiments on this topic. I would also strongly encourage the authors to undertake or promote the study of how realistic these model organisms are, to grow the applicability of this relatively new area.
> >
> > I also appreciate the new results on MMLU and thank you for your time and effort producing them in the rebuttal period.

---

> > > ### Author Response · Authors · 2025-04-08
> > >
> > > Thank you, we really appreciate your good-faith engagement. We will add further discussion on the points you raised.

---

### Official Review · Reviewer_XiFg · 2025-03-14

**Overall Recommendation:** 4

**Summary:**

This paper introduces Model organisms which are language models that are trained to imitate AI systems with human capabilities via circuit breaking. This method of training is more robust to elicitations than simple password-locked training approach. On the tasks of MCQA and code generation, they are able to show that this approach works better. For the code generation task, only fine-tuning approach can elicit the abilities of the model.

**Claims And Evidence:**

Yes

**Essential References Not Discussed:**

n/a

**Experimental Designs Or Analyses:**

Yes the experiments have been carefully designed and thorough.

**Methods And Evaluation Criteria:**

Yes

**Other Comments Or Suggestions:**

- Some theoretical analysis on why different elicitation methods are suited for different tasks eg. here code generation vs hazardous knowledge could be interesting to understand the workings could be a future work direction

**Other Strengths And Weaknesses:**

Strengths
- This area of study is still fairly nascent and this work makes a valuable contribution by demonstrating what kinds of techniques are most useful for eliciting the hidden abilities of models.
- By proposing a novel model organism approach of circuit-breaking training after password-locked training, they show increased robustness to elicitation on 2 tasks.
- The discussion section is very informative, and several interesting points made.

Weakness
- Only 2 tasks studied.

**Questions For Authors:**

- Why did you choose these 2 specific tasks, any thoughts on how this would scale to other tasks?

**Relation To Broader Scientific Literature:**

Capability evaluation is an important assessment with prior works illustrating the sandbagging phenomenon where models are made to strategically underperform to pass regulatory bodies. This work shows how elicitation techniques based on fine-tuning are more effective. When there is not much data available, a combination of in-filling, steering etc works best.

**Theoretical Claims:**

n/a

---

> ### Author Rebuttal · Authors · 2025-04-01
>
> Thank you for taking the time to review our paper. We’re glad that you appreciated our contribution to the capability evaluation and sandbagging literature, with our evaluation of techniques for eliciting hidden capabilities, and the introduction of a more robust model organism.
>
> > Why did you choose these 2 specific tasks, any thoughts on how this would scale to other tasks?
>
> We chose the WMDP benchmark as it has been used in prior work on sandbagging, and is a clear demonstration of hazardous knowledge assessed in dangerous capability evaluations.
>
> Because WMDP is an MCQA setting, we investigated the APPS setting for code-generation, as it is more open-ended and realistic. Code generation is also relevant to dangerous capability evaluations, in the context of, for example, cyber-offensive capabilities.
>
> In line with your feedback, we have extended our results to a third benchmark — MMLU.
> For password-locked models, the results are essentially consistent with those for WMDP.
> For Mistral-base and Mistral-instruct circuit-broken models, the effect of the circuit-breaking seems stronger than what we observed for WMDP (i.e., the circuit-broken models are even more robust than regular password-locked models).
> However, Gemma-instruct circuit-broken seems less robust to elicitation than its password-locked counterpart. As we noted in the paper (lines 305 - 310), circuit-breaking is quite sensitive to training hyperparameters, which may explain this result.
> In accordance with our initial results, anti-refusal training and SFT are the most effective elicitation methods.
>
>
> Below we provide the full results for MMLU after applying each elicitation technique on the password-locked and circuit-broken model organisms for each model. For N-shot prompting, we use N=1, 3, 5 and 10 shots, and present the best result. When combining steering with other techniques, we use persona steering.
>
> ## MMLU Results
> ### Mistral-base
> |Model organism|Baseline without password|Baseline with password|N-shot|Prefill|Prefill + N-shot|Concept steering|Persona steering|Prefill+S|Prefill + Steering + N-shot|Anti-refusal training|SFT|
> |-|-|-|-|-|-|-|-|-|-|-|-|
> |*Password-locked*|11.5|62.0|60.4|59.2|62.1|51.7|56.2|59.6|60.6|51.0|**65.6**|
> |*Circuit-broken*|1.1|62.8|27.2|0.0|30.0|9.3|0.0|0.0|28.5|52.5|**64.0**|
>
>
> ### Mistral-instruct
> |Model organism|Baseline without password|Baseline with password|N-shot|Prefill|Prefill + N-shot|Concept steering|Persona steering|Prefill+S|Prefill + Steering + N-shot|Anti-refusal training|SFT|
> |-|-|-|-|-|-|-|-|-|-|-|-|
> |*Password-locked*|10.0|59.4|57.2|58.9|57.1|43.8|26.0|58.5|57.8|53.0|**64.1**|
> |*Circuit-broken*|0.0|59.3|4.4|1.8|57.1|0.3|0.0|11.3|56.9|55.7|**61.1**|
>
>
> ### Gemma-instruct
> |Model organism|Baseline without password|Baseline with password|N-shot|Prefill|Prefill + N-shot|Concept steering|Persona steering|Prefill+S|Prefill + Steering + N-shot|Anti-refusal training|SFT|
> |-|-|-|-|-|-|-|-|-|-|-|-|
> |*Password-locked*|6.6|72.5|28.9|30.2|72.1|38.5|72.5|34.2|72.8|71.0|**76.3**|
> |*Circuit-broken*|6.1|75.5|7.7|67.1|72.1|71.1|72.2|66.7|72.0|70.9|**75.5**|
>
> > Some theoretical analysis on why different elicitation methods are suited for different tasks eg. here code generation vs hazardous knowledge could be interesting to understand the workings could be a future work direction
>
> We agree that this would be interesting! However, providing these analyses with sufficient rigour, especially given the multiple elicitation methods included in our paper, requires effort that is out of scope for the current work.
> Regardless, we agree with you that this is a fruitful topic for future research, and therefore we hope to provide some preliminary hypotheses in lines 360-372 (right column). We believe our paper is a stepping stone towards future theoretical research on elicitation methods.
>
> An interesting and important insight from our work is that hidden capabilities can be embedded within specific layers of a model through mechanisms such as LoRA adapters and/or circuit-breaking. This suggests that evaluators who fine-tune models at specific layers—common practice with LoRA—may remain vulnerable to the persistence of hidden capabilities. Concretely, circuit-broken model organisms can be trained in a way that fine-tuning only on the latest 40% of layers does not elicit performance. While this point was accidentally excluded from the original submission, it has now been incorporated into the revised manuscript.
>
> ----
> Thanks again for your help and feedback!

---

> > ### Comment · Reviewer_XiFg · 2025-04-06
> >
> > Dear authors, thank you for clarifying my question and providing results on MMLU! I will retain my acceptance score.

---

### Decision · Program_Chairs · 2025-05-01

**Decision:**

Accept (poster)

**Comment:**

This paper investigates techniques for eliciting hidden capabilities in language models. To do so, the authors train "model organisms" with intentionally hidden behaviours, using both password-locking and a circuit-breaking approach. The paper evaluates a variety of elicitation methods (prompting, activation steering, and fine-tuning) on multiple-choice QA and code generation tasks. Results show that the circuit-breaking approach is more robust to elicitation, and while the elicitation technique efficiency varies between tasks and models, fine-tuning generally outperforms other approaches, especially on the more difficult code task, where it is the only method able to reliably recover the model’s latent capabilities.

This paper tackles an important and timely problem: how to evaluate elicitation techniques for uncovering hidden capabilities in LLMs, which is a key concern for ensuring safe deployment. While the experiments cover only a small number of tasks (a third was added in the rebuttal), the main concern raised by reviewers is whether the relatively small models used (e.g. 7B) are indicative of real-world LLMs (100s of billions or trillions of parameters) that will see widescale deployment. This is a valid limitation, which is acknowledged by the paper.  Ultimately, though, I am in agreement that the paper's contribution is strong enough to stand on its own at present and that the work provides a valuable foundation for future studies on more capable models.